# Preparation of Amine-Modified Cu-Mg-Al LDH Composite Photocatalyst

**DOI:** 10.3390/nano12010127

**Published:** 2021-12-30

**Authors:** Qining Wang, Quanwang Yan, Yu Zhao, Jie Ren, Ning Ai

**Affiliations:** 1National Demonstration Center for Experimental Chemistry and Chemical Engineering Education, Zhejiang University of Technology, Hangzhou 310014, China; wqn989@zjut.edu.cn; 2College of Chemical Engineering, Zhejiang University of Technology, Hangzhou 310014, China; yanquanwang0@163.com (Q.Y.); zhaoyu11@sinochem.com (Y.Z.); 3School of Biology and Chemical Engineering, Jiaxing University, Jiaxing 314001, China; renjie@zjxu.edu.cn; 4Sinochem Lantian Trading Co., Ltd., Hangzhou 310051, China

**Keywords:** layered double hydroxide, Cu^2+^, amine modification, carbon dioxide, capture, photocatalytic conversion

## Abstract

Cu-Mg-Al layered double hydroxides (LDHs) with amine modification were prepared by an organic combination of an anionic surfactant-mediated method and an ultrasonic spalling method using *N*-aminoethyl-*γ*-aminopropyltrimethoxysilane as a grafting agent. The materials were characterized by elemental analysis, XRD, SEM, FTIR, TGA, and XPS. The effects of the Cu^2+^ content on the surface morphology and the CO_2_ adsorption of Cu-Mg-Al LDHs were investigated, and the kinetics of the CO_2_ adsorption and the photocatalytic reduction of CO_2_ were further analyzed. The results indicated that the amine-modified method and appropriate Cu^2+^ contents can improve the surface morphology, the increase amine loading and the free-amino functional groups of the materials, which were beneficial to CO_2_ capture and adsorption. The CO_2_ adsorption capacity of Cu-Mg-Al N was 1.82 mmol·g^−1^ at 30 °C and a 0.1 MPa pure CO_2_ atmosphere. The kinetic model confirmed that CO_2_ adsorption was governed by both the physical and chemical adsorption, which could be enhanced with the increase of the Cu^2+^ content. The chemical adsorption was suppressed, when the Cu^2+^ content was too high. Cu-Mg-Al N can photocatalytically reduce CO_2_ to methanol with Cu^2+^ as an active site, which can significantly improve the CO_2_ adsorption and photocatalytic conversion.

## 1. Introduction

Uncontrolled burning of fossil fuels, since the industrial revolution, has not only caused serious energy consumption problems, but also made global warming an urgent problem for human sustainable development due to the sharp increase of CO_2_ concentration in the atmosphere [1,2,3]. For this reason, capturing CO_2_ from sources and converting it to carbon-containing fuels or high-value non-fuel chemicals are currently relatively attractive solutions to avoid extreme climate change [4,5]. As an extensively studied family of solid adsorbent, layered double hydroxides (LDHs) can realize the coupling of CO_2_ capture and utilization. Because LDHs materials have attractive physiochemical properties, such as characteristic layered structures with large surface areas, specific “memory effects”, high anion exchange capacity, precise morphological controllability, facileness to synthesize, and low costs [6,7,8,9,10,11].

Despite these positive properties, unactivated LDHs exhibit unstable structures and relatively low CO_2_ adsorption capacities in the range of 0.1 to 0.49 mmol·g^−1^ [12,13,14]. It is easy for substrates or reactants to penetrate the interlayer space, given the more defective and porous nature of the interlayer space for unactivated LDHs. It was proposed that the high stability comes from the less defective particle surface and the highly dense interlayer structure and manifested as a considerably higher crystallinity and a layer charge density [15]. Bond lengths, cell angles, electronic structure of cations, and the Mulliken population of clusters play a significant role in relative stability of LDHs [16]. In order to improve the structural stability and CO_2_ capture capacities, many attempts have been made, such as various synthesis approaches [17,18], modification of composition [19,20,21], morphology controlling [22,23], doping LDHs with alkali metal [24], and the formation of hybrid materials [25]. Enhanced capacities can be obtained by intercalating amino acids or amino silanes into stable layers and reacting with hydroxyl groups on laminates to achieve an amine loading. In light of the promising amine loading of intercalated LDHs, our research group proposed an amine-modified method [26,27], namely ultrasonic spalling [28] after co-precipitation [29], to further improve the adsorption performance of CO_2_.

LDHs, as excellent photocatalytic materials [30,31], have been applied in CO_2_ capture and conversion into small-molecular organic compounds such as CH_4_, CH_3_OH, and C_2_H_5_OH by photocatalytic reaction [32,33,34,35]. A series of LDHs can serve as sacrificial hard templates by host–guest structural design and assembly [36], of which the characteristic band gap can be adjusted with a wide choice of interlayer anions and metal cations [37]. LDHs, integrated with metals-bearing, high electron storage capacity and electron transportation efficiency [38,39], have great potential to enhance photocatalytic activity on the basis of the high adsorption capacity of CO_2_. Hong et al. [40] determined that the Mg-Al LDH, due to its excellent CO_2_ adsorption capacity, contributes to a remarkably high photocatalytic reduction efficiency of CO_2_ to CH_4_ in the presence of the carbon nitride (C_3_N_4_) photoabsorber and the Pd cocatalyst. The results show that the CH_4_ production of the Mg–Al LDH is 2.6 times that without the LDH coating and better than that of the Zn-Al LDH, the Ni-Al LDH, and the Zn-Cr LDH. Research on LDHs has extended from simple bicationic to tricationic or tetracationic multiphased species [41,42], of which the photocatalytic performances vary with the doping of different metal cations. Ahmed et al. [43,44] reported for the first time that LDHs can be applied as photo-catalysts to convert gaseous CO_2_ under UV-visible light using hydrogen. Compared with CO formed by Zn-Al LDHs, methanol is the major product formed by the inclusion of Cu sites using Zn-Cu-Ga LDHs, which suggested that the specific interaction of Cu sites with CO_2_ enables the formation of methanol by coupling with protons and photogenerated electrons. The efficient photocatalytic conversion of CO_2_ into methanol can also be observed over Zn-Cu-Ga LDHs by Morikawa et al. [45] in the preparation of reverse fuel cells.

The inclusion of Cu sites in LDH layers can improve the methanol selectivity; however, the Cu^2+^ content has a great influence on the CO_2_ adsorption properties and catalytic activity of LDHs, which is ascribed to the distortion induced by the Jahn–Teller effect of Cu^2+^ [46,47,48]. Hence, the preparation of Cu-Mg-Al LDHs with excellent CO_2_ adsorption performance is of great significance. In this regard, the aims of this work were to prepare, characterize and evaluate the CO_2_ adsorption capacity and photocatalytic conversion performance of Cu-Mg-Al LDHs with an appropriate Cu^2+^ content. The as-synthesized materials were prepared by an amine-modified method, while N-aminoethyl-γ-aminopropyltrimethoxysilane as a grafting agent was then characterized to examine its structures and morphologies. The study focused on the effects of the amine-modified method and the Cu^2+^ content on the CO_2_ adsorption performance and the kinetics of the adsorption on Cu-Mg-Al LDHs. Substituting part of Mg^2+^ with Cu^2+^ during the preparation of LDHs made it possible to improve the CO_2_ photocatalytic conversion performance.

## 2. Materials and Methods

### 2.1. Materials

All of the chemicals were analytical-grade reagents and used as received without any further purification. Copper nitrate trihydrate (Cu(NO_3_)_2_·3H_2_O; ≥98%), magnesium nitrate hexahydrate (Mg(NO_3_)_2_·6H_2_O; ≥98%), aluminum nitrate nonahydrate (Al(NO_3_)_3_·9H_2_O; ≥98%), sodium dodecyl sulfonate (C_12_H_25_SO_3_Na; ≥97%), and *N*-aminoethyl-*γ*-aminopropyltrimethoxysilane (C_8_H_22_N_2_O_3_Si; ≥95%) were purchased from Aladdin Industrial Inc. (Shanghai, China). Sodium hydroxide (NaOH; ≥98%) and toluene (C_7_H_8_; ≥99.5%) were obtained from Xilong Scientific Co., Ltd. (Guangzhou, China). Ethanol (C_2_H_5_OH; ≥99.7%) was supplied from Anhui Ante Food Co., Ltd. (Suzhou, Anhui, China). Additionally, deionized water was used to formulate the solution throughout the experiment.

### 2.2. Preparation of Amine-Modified Cu-Mg-Al LDHs

Cu-Mg-Al LDHs materials with a constant M^II^/M^III^ mole ratio of 3 were prepared by an amine-modified method while using Cu(NO_3_)_2_·3H_2_O, Mg(NO_3_)_2_·6H_2_O, and Al(NO_3_)_3_·9H_2_O as precursors.

The first-stage process was to synthesize the primary amine-modified LDHs by co-precipitation. About 1.95 g of Al(NO_3_)_3_·9H_2_O, a certain amount of Cu(NO_3_)_2_·3H_2_O, and Mg(NO_3_)_2_·6H_2_O with a Al^3+^/Cu^2+^/Mg^2+^ molar ratio of 1:x:3−x were dissolved in 50 mL deionized water (solution A). Then, 2.83 g of sodium dodecyl sulfonate (DS) and 6.94 g of *N*-aminoethyl-*γ*-aminopropyltrimethoxysilane (N) were dissolved in a mixture of 100 mL deionized water and 50 mL ethanol (solution B). The M^II^:M^III^:DS:N substrate mole ratio was 3:1:2:6. Solution A was then added dropwise to solution B at 70 °C. The mixture was stirred, before the pH value was stabilized at 10 ± 0.1 by adding a 1.0 M NaOH solution. After adding solution A, the mixture was then aged for another 4 h with the stirring and temperature maintained. The precipitated solids were isolated by filtration, washed with deionized water and dried under vacuum overnight. Varying the Cu^2+^ content, three samples were produced with the Al^3+^:Cu^2+^:Mg^2+^ molar ratios of 1:0.75:2.25, 1:1:2, and 1:1.25:1.75. The corresponding samples were labeled as 0.75Cu-Mg-Al DS/N, Cu-Mg-Al DS/N, and 1.25Cu-Mg-Al DS/N. xCu-Mg-Al DS/N was used as a general designation for all three primary amine-modified LDHs. In addition, non-modified, pure Cu-Mg-Al LDHs was named Cu-Mg-Al DS.

The second stage process was to remodify xCu-Mg-Al DS/N by ultrasonic spalling. The dispersion of 1.0 g of xCu-Mg-Al DS/N in 100 mL toluene was subjected to sonication for 5 h with an ultrasonic machine (JY 92-IIN, SCIENTZ, Ningbo, China). The mixture was stirred at 70 °C for 20 h after adding 4 g of *N*-aminoethyl-*γ*-aminopropyltrimethoxysilane. The obtained materials were filtered, washed with deionized water and dried under vacuum overnight. The corresponding samples after ultrasonic spalling were labeled as 0.75Cu-Mg-Al N, Cu-Mg-Al N, and 1.25Cu-Mg-Al N. xCu-Mg-Al N was used as a general designation for all three amine-modified LDHs.

### 2.3. Characterization

The quantitative analyses of carbon, hydrogen, nitrogen, and sulfur contents in the LDHs were performed by an elemental analyzer (VATIO MACRO CUBE; Elementar, Langenselbold, Germany). The structures and compositions of the LDHs were identified by X-ray powder diffraction spectrometer (X’Pert Pro; PNAlytical, Almelo, The Netherlands) with Cu Kα radiation (λ = 0.1541 nm) at 40 kV and 40 mA. The patterns were collected with diffraction angels (2*θ*) ranging from 2° to 65°. The surface morphologies of the LDHs were observed by a scanning electron microscope (VEGA3; Tescan, Brno, Czech Republic) with an applied voltage of 10 kV. The surface functional groups of the LDHs, under vacuum and dried at 70 °C for 4 h prior to analysis, were detected on a Fourier-transform infrared spectrometer (TENSOR II; Bruker, Karlsruhe, Germany) in a wavelength region of 4000 to 400 cm^−1^ and in the transmission mode with a resolution of 4 cm^−1^ for 64 scans. The thermal decompositions of the LDHs were evaluated by a thermogravimetric (TG) analyzer (TG209F3; Netzsch, Luxemburg, Germany) with a N_2_ flow of 30 mL·min^−1^ at a heating rate of 10 °C·min^−1^. The surface elemental compositions of the LDHs were detected by an X-ray photoelectron spectrometer (ESCALAB 250Xi, Thermo Fischer, Waltham, MA, USA) using monochromatic Al Kα as the X-ray source (1486.6 eV). The binding energy was calibrated by using the C 1s peak at 284.6 eV as a reference.

### 2.4. CO_2_ Adsorption Capacity and Sequential Adsorption-Regeneration Cycles

The CO_2_ adsorptions of the amine-modified Cu-Mg-Al LDHs were tested using the TGA method [29]. In a typical test, approximately 10 mg of each sample were loaded into a ceramic microbalance and then pretreated from room temperature to 140 °C at a heating rate of 15 °C·min^−1^ in a flow of N_2_ for 30 min to remove any remaining CO_2_ from synthesis and storage in atmosphere. The temperature was then brought back to the desired adsorption temperature at a cooling rate of 10 °C·min^−1^ for 30 min. After the sample weight became stable, the gas input was switched from N_2_ to pure CO_2_ and then held isothermally for 180 min until the dynamic adsorption equilibrium. The CO_2_ adsorption capacity was computed according to the mass variation of the sample in the CO_2_ atmosphere.

In order to evaluate the cyclic adsorption property, the adsorption-regeneration cycle was repeated three times as follows: after pretreatment, the CO_2_ adsorption was carried out at 30 °C in a pure CO_2_ atmosphere for 150 min, and regeneration was carried out at 140 °C in the pure N_2_ atmosphere for 30 min.

### 2.5. Photocatalytic CO_2_ Reduction Test

The photocatalytic performance of the sample was carried out in a 500 mL cylindrical airtight reactor (CEL-HXF300, Au-light, Beijing, China) with a 2 cm thick high-transmitting quartz glass on the top. One gram of the sample was spread evenly onto the bottom of the reactor. CO_2_ was blown into the reactor to ensure pressure stability at 0.1 MPa, and then, the hydrogen pressure was stable at 0.4 MPa. The reactor was illuminated for 1 h by a PerkinElmer 300 W Xe lamp as a simulated light source. After the reaction was over, 30 mL of the gas mixture extracted from the reactor by a syringe were analyzed by a gas chromatograph (GC, GC9790, Fuli, Taizhou, China) equipped with a flame ionization detector (FID) and Porapak Q as the column chromatograph.

## 3. Results and Discussion

### 3.1. Characterization Results of Cu-Mg-Al LDHs

#### 3.1.1. Elemental Analysis

The formula for the intercalated molecules was calculated based on the weight percentages of the carbon, hydrogen, nitrogen, and sulfur in the Cu-Mg-Al LDHs, and the amount of the amine loading was calculated as (weight of N content × 1000)/14, as shown in Table 1. Compared with xCu-Mg-Al DS/N, a decrease in the S content of xCu-Mg-Al N indicated that the amine-modified method can exfoliate the intercalated anionic surfactant during ultrasonic spalling. Observed from the formula for intercalated molecules, the increment of amino silanes was greater than the decrement of the anionic surfactant, which indicated that amino silanes partially replaced the anionic active sites and condensed with the surface hydroxyl groups on the laminates.

The amine loading of xCu-Mg-Al N after amine modification was increased significantly, and the more Cu^2+^ content in the sample, the greater the increase. As Cu^2+^ continuously replaces Mg^2+^ in the layer, the charge density of the laminates decreased, which can be attributed to the fact that the charge population of copper was smaller than that of magnesium [49]. Simultaneously, the electrostatic interactions between the cations of laminates and the interlayer anions weakened. Consequently, resulting in the interlayer spacing being expanded implied more conducive to the intercalation of the grafted amino silanes.

#### 3.1.2. XRD Analysis

Figure 1 shows the XRD patterns of xCu-Mg-Al DS/N and xCu-Mg-Al N. In all samples, the nonbasal reflections (006) at a 2*θ* of about 5° and (110) at a 2*θ* of about 60° were preserved, indicating that the structure of layers was conserved [29,50]. xCu-Mg-Al DS/N and xCu-Mg-Al N exhibited similar patterns from a rough look. However, each sample had a different intensity of the reflections at the peaks from a careful observation.

According to the Bragg’s Law (2*d*sin*θ* = *nλ*, where *θ* is the diffraction half-angle and *n* is the diffraction order) and subtracting the average thickness of the laminate by 0.49 nm [50], the interlayer spacing of Cu-Mg-Al N was calculated to be 3.58 nm by the (003) diffraction peak located at 2.17°. Compared with 1.98 nm of the non-modified Cu-Mg-Al DS and 3.09 nm of the primary amine-modified Cu-Mg-Al DS/N, the enlarged interlayer spacing governed by the size of the anions showed that the amine-modified method can further improve amino silanes grafting.

The intensity and sharpness of the (003) peak reduced gradually with the progress of amine modification and the increase of the Cu^2+^ content, and the overlapping of the peaks corresponding to the (015) and (018) reflections occurred. Combined with the amine loading data of the elemental analysis in Table 1, it was shown that the crystallinity of the particles decreased after augmenting amino silanes grafting sites [50].

It is noticeable that as the Cu^2+^ content increased, the (003) peak of 1.25Cu-Mg-Al N detectably shifted toward a large angle, and the reflection in the 2*θ* range of 15° to 30° became narrow, which is mainly due to the distortion induced by the Jahn–Teller effect of Cu^2+^. With an increase of the Cu^2+^ content in the layer, this distortion weakened the hydrogen bonding and the electrostatic interactions between the host layer and the guest. The absolute value of the binding decreased, and the chemical stability of the system decreased as well. Correspondingly, the interlayer spacing of 1.25Cu-Mg-Al N was shortened to 3.17 nm, which displayed that the host layer and the guest needed to be close to each other in order to increase the interaction force [51] and how seriously the stability of the system declined.

#### 3.1.3. Microstructure Observation

The morphologies of xCu-Mg-Al N and xCu-Mg-Al DS/N were further confirmed by SEM micrographs. The particle size and the surface morphology of xCu-Mg-Al N (Figure 2a–c) changed dramatically by the amine-modified method when compared to those of xCu-Mg-Al DS/N (Figure 2d–f). The surface of xCu-Mg-Al DS/N had a plate-like shape. The diameters of xCu-Mg-Al N ranged from 20 to 50 μm, which were averagely smaller than the lateral sizes of xCu-Mg-Al DS/N, accompanied by a completely loading with small particles on the surface.

The Cu^2+^ content also had a greater influence on the surface morphology of xCu-Mg-Al N. Compared with the chunky grafts aggregated on the surface of 0.75Cu-Mg-Al N (Figure 2a), dispersing Cu-Mg-Al N (Figure 2b) into granular grafts was more favorable for CO_2_ trapping and adsorption to some extent. The surface of 1.25Cu-Mg-Al N (Figure 2c) was abnormally smooth with remarkably increased agglomeration, which speculated that the surface grafts were polymerized to maintain the system stability. The high Cu^2+^ content introduced more defects into the system and resulted in excessive distortion by the Jahn–teller effect, which was consistent with the XRD pattern results and supported our inference.

#### 3.1.4. FTIR Analysis

FTIR analysis was further conducted to identify functional groups on the surface of LDHs. Figure 3 shows the variations in the FTIR spectra of xCu-Mg-Al DS/N and xCu-Mg-Al N. For xCu-Mg-Al DS/N, the bands at 3250 cm^−1^ (stretching vibration of –NH_2_ groups) and 1588 cm^−1^ (bending vibration of –NH_2_ groups) were assigned to the characteristic absorption bands of amino silanes; meanwhile, apparent peaks at 1185/1140 cm^−1^ (stretching vibration of –SO_3_) were observed, corroborating that *N*-aminoethyl-*γ*-aminopropyltrimethoxysilane and sodium dodecyl sulfonate were grafted to the LDH layer in the form of salt after the primary amine modification.

For xCu-Mg-Al N, a further improvement of amino silanes grafting after amine modification was confirmed by the presence of new peaks at 1311 cm^−1^ for the stretching vibration of N–C and 1114 cm^−1^ for the stretching vibration of Si–O–C. The weakening of the –SO_3_ stretching vibration after amine modification may be due to the exfoliation of LDH laminates by ultrasonication. The intensities of those characteristic bands were substantially enhanced by felicitously increasing the Cu^2+^ content. In other words, the surface functional groups of Cu-Mg-Al N were similar to those of 0.75Cu-Mg-Al N, but the surface grafts of Cu-Mg-Al N were denser. Previous predictions are consistent with the results of the microstructure observation and the elemental analysis.

For 1.25Cu-Mg-Al N, the intensities of the –OH stretching vibration in the range of 3300 to 3500 cm^−1^ were significantly weaker than those of 0.75Cu-Mg-Al N and Cu-Mg-Al N, indicating that the amine loading of 1.25Cu-Mg-Al N was increased obviously. What’s more, the bending vibration of –NH_2_ groups at 1588 cm^−1^ overlapped with the stretching vibration of water molecules in the same IR region of 1.25Cu-Mg-Al N, which may be due to the fact that amino silanes polymerized on the surface and the content of the free-amino groups was greatly reduced.

#### 3.1.5. Thermal Analysis

TG study was carried out to investigate the thermal stability of the samples. As it is presented in Figure 4, the Cu-Mg-Al LDHs underwent three stages of mass loss. The first stage occurring below 150 °C may be attributed to the removal of the interlayer water and the physically adsorbed CO_2_ [28]; the second stage appearing in the range from 150 to 300 °C mainly may be ascribed to substrate dehydroxylation [52]; the third stage detected at temperatures over 300 °C may be assigned to the carbon chain decomposition of anionic surfactants and amino silanes [29].

As seen from Table 2, the weight loss ratio of xCu-Mg-Al N in the second stage was obviously lower than that of xCu-Mg-Al DS/N, which indicated that the free-hydroxyl groups on the surface of the material decreased after being remodified by ultrasonic spalling. It was inferred that the amine-modified process with more grafted amino silanes can promote the condensation between the amino silanes and the surface hydroxyls on the laminates. The weight loss ratios in the third stage were significantly different. The weight loss ratios of 0.75Cu-Mg-Al N and Cu-Mg-Al N were lower than those of the corresponding primary amine-modified LDHs, while the weight loss ratio of 1.25Cu-Mg-Al N was higher than that of the corresponding primary amine-modified LDH. Combined with the formula for the intercalated molecules in Table 1, the former was attributed to the fact that the increase of the amine loading was similar to that of the anionic surfactant removal and the molecular weight of the amino silane was obviously smaller than that of the anionic surfactant, while the latter can be related to the fact that the increment of the amino silanes was much greater than the decrement of the anionic surfactant.

It was noted that the weight loss ratios of xCu-Mg-Al N decreased in the first two stages accompanied by the Cu^2+^ content increase. Associated with the formula for the intercalated molecules in Table 1, the first stage may be interpreted that the interlayer anionic sites were gradually replaced by amino silanes, of which the hydrophilicity was weaker than that of the anionic surfactants. The second stage may be due to the consumption of the free-hydroxyl groups promoted by Cu^2+^. Furthermore, one obvious difference is that the weight loss ratio of 1.25Cu-Mg-Al N increased on the third stage, and it was speculated that part of amino silanes on the surface of sample may have been polymerized.

#### 3.1.6. XPS Analysis

The effect of the Cu^2+^ content on the aminated functional groups in the amine modification process was further investigated via XPS analysis. N 1s spectra were recorded in Figure 5 to analyze the surface compositions of the samples. The high-resolution N 1s spectra of xCu-Mg-Al N (Figure 5a–c) can be deconvoluted into two peaks at ca. 399.3 eV (peak 1) and 401.3 eV (peak 2) binding energies, which corresponded to free amine and protonated amines, respectively [53], and that of xCu-Mg-Al DS/N (Figure 5d–f) was slightly offset. As seen from Figure 5, the contents of the free amines and protonated amines on the surface of xCu-Mg-Al N were more than those of xCu-Mg-Al DS/N, which indicated that increasing the surfactant removal rate significantly improved the amine loading on the surface of the samples. As the addition of Cu^2+^ increased, a higher concentration of free amines was shown, which revealed that there were comprehensive amino group active sites readily available for CO_2_ adsorption.

The XPS spectra of the Cu 2p region were utilized to better verify the chemical states of Cu species in the samples, as shown in Figure 6. Two main peaks were observed at binding energies of ca. 952 and 932 eV. In a nutshell, the peak at the binding energy of ca. 932 eV can be deconvoluted into two sets of peaks, in which the peaks at ca. 934.3 eV (peak 3) and 932.3 eV (peak 1) can be assigned to 2p_3/2_ of Cu^2+^ and 2p_3/2_ of either Cu^0^ and/or Cu^+^, respectively [54]. The former peak can be assigned to Cu^2+^ species, and the latter may be related to another state where Cu ions coordinate with Mg and Al ions in a spinel-like species [55]. The other two peaks at ca. 954.4 eV (peak 4) and 952.1 eV (peak 2) deconvoluted by the peak at the binding energy of ca. 952 eV can correspond to 2p_1/2_ of Cu^2+^ and 2p_1/2_ of either Cu^0^ and/or Cu^+^, respectively. Unlike the others, the area of peak 4 for 1.25Cu-Mg-Al N was obviously higher than those of the others, even 0 for Cu-Mg-Al N, which revealed that the disappearance of 2p_1/2_ of Cu^2+^ at the high energy side was better in promoting CO_2_ adsorption.

### 3.2. CO_2_ Adsorption Performance

#### 3.2.1. Effect of the Amine Loading

The amine loading through the availability of basic amine groups mainly determines the CO_2_ adsorbents of solid-supported amine materials [28,56]. Figure 7 displays the relationship between the CO_2_ adsorption capacity of the samples and the amine loading according to elemental analysis at 30 °C and a 0.1 MPa pure CO_2_ atmosphere. As the Cu^2+^ content increased, the total amine loading of each sample increased, whereas the correlation of CO_2_ adsorption capacity with the Cu^2+^ content increase was not positive. The CO_2_ adsorption capacities of 0.75Cu-Mg-Al N and Cu-Mg-Al N both increased by over 1 mmol·g^−1^ when compared to those of the corresponding primary amine-modified LDHs, indicating that the amine-modified method can significantly improve the CO_2_ adsorption performance of the material. In contrast, 1.25Cu-Mg-Al N exhibited a very disheartening CO_2_ adsorption capacity down to 0.70 mmol·g^−1^, which was even lower than that of 1.25Cu-Mg-Al DS/N. It suggested that the amine loading was not the only factor affecting the CO_2_ adsorption performance of the material [52]. The phenomenon of the inhibited CO_2_ adsorption can be explained by the increase in the interlayer spacing, caused by the decrease in the charge density of the laminates due to overdoped Cu^2+^, promoting the intercalation of amino silanes and weakening the stability of the materials; meanwhile, the distortion induced by the Jahn–Teller effect led to the mass polymerization of amino silanes on the sample surface.

#### 3.2.2. Effect of the Temperature

Figure 8 shows the CO_2_ adsorption capacities of xCu-Mg-Al N and xCu-Mg-Al DS/N at three different temperatures at a 0.1 MPa pure CO_2_ atmosphere. The CO_2_ adsorption capacities of all samples decreased with the increasing temperature, and the highest one was up to 1.82 mmol·g^−1^ for Cu-Mg-Al N at 30 °C. This is due to the reaction being exothermic and the increase in the temperature being unfavorable to the adsorption. Nevertheless, the adsorption capacities of 0.75Cu-Mg-Al N and Cu-Mg-Al N at higher temperatures in the range of 1.4 to 1.7 mmol·g^−1^ were much larger than those of the primary amine-modified samples, which suggested that Cu-Mg-Al LDHs prepared by the amine-modified method have the potential to be used in the adsorption of flue gas from fossil fuel-based thermal power plants. Simultaneous with the temperature of the reactor rising during the photocatalysis process, this conclusion provides the theoretical support of adsorption for further applications of Cu-Mg-Al LDHs to the CO_2_ photocatalytic conversion.

For a comparison, the CO_2_ adsorption capacities for the amine-modified LDHs, as reported in the literature, are listed in Table 3, where it can be seen that the CO_2_ adsorption capacity of Cu-Mg-Al N was either better or comparable with those of many other materials investigated.

#### 3.2.3. Effect of Sequential Adsorption-Regeneration Cycles

The stable recycling performance of adsorbents is a key research content in industrial process. Figure 9 represents the cyclic adsorption-regeneration performance of Cu-Mg-Al N, which showed a better adsorption property in previous experiments. The deactivation rate of the adsorbent was calculated by dividing the difference in the adsorption capacities between the 1st and 2nd cycles by adsorption capacity after the 1st cycle. The Cu-Mg-Al N adsorbent exhibited a slight decrease in the deactivation rate of only 8% from the initial stage to the 3rd cycle, which may be attributed to the formation of stable semicarbazide deposited on the adsorbent by the dehydration of a small fraction of CO_2_ with the amino group at high temperatures [53,57]. This result revealed that Cu-Mg-Al N displayed excellent adsorption performance stability throughout three sequential adsorption-regeneration cycles, which is also an essential criterion for practical applications.

#### 3.2.4. Adsorption Kinetics

Adsorption kinetics analysis is an important index to evaluate the efficiency of adsorbents [58]. Pseudo-first-order, pseudo-second-order, and double-exponential models were applied to the experimental data on Cu-Mg-Al LDHs through nonlinear regression. The kinetic parameters are presented in Table 4, and the fitting curves under different kinetic models were compared with the experimental data points in Figure 10. From the comparison results, it can be seen that double-exponential model displayed a comparatively good fit with the CO_2_ adsorption process of Cu-Mg-Al LDHs.

Loganathan et al. [59] found that the excellent fit obtained using the double-exponential model resulted from its capacity to account for both the physical and chemical adsorptions of CO_2_ on an adsorbent. The factors A1 and A2 of the double-exponential model in Table 4 represent the maximum adsorption capacity, while k1 and k2 stand for the kinetic rate constants of the two adsorption sites. The lower value of k1 corresponds to the physisorption site through intra-particle diffusion, whereas the higher value of k2 accounts for the chemisorption site through strong surface reactions. From the A1 and A2 values of xCu-Mg-Al N and xCu-Mg-Al DS/N, it can infer that CO_2_ adsorption takes place via chemisorptive and physisorptive interactions. The remarkable improvements in the overall adsorption performances of 0.75Cu-Mg-Al N and Cu-Mg-Al N were assigned to the obvious enhancements in both physisorption and chemisorption after amine modification. The suppressed chemisorption of 1.25Cu-Mg-Al N caused by the agglomeration of particles was predominant in the unimproved adsorption performance, although physisorption was enhanced.

### 3.3. Photocatalytic CO_2_ Reduction

The influences of different photocatalysts on the catalytic activity were investigated by CO_2_ reduction under a solar simulator, and the relevant parameters of gas chromatography analysis are shown in Table 5. Compared with Mg-Al N without the Cu^2+^ doping, methanol could be obtained by the CO_2_ reduction with both Cu-Mg-Al DS/N and Cu-Mg-Al N as photocatalysts, which indicated that Cu^2+^ was the active site of photocatalytic reaction, and demonstrated the potential of amine-modified Cu-Mg-Al LDHs in the field of photocatalytic reduction to methanol.

The methanol yield of Cu-Mg-Al N was significantly higher than that of Cu-Mg-Al DS/N, that is, more photocatalytic active sites provided by Cu-Mg-Al N, which resulted in the higher photocatalytic conversion efficiency of Cu-Mg-Al N [60]. It suggested that amine modification not only conspicuously improves the CO_2_ adsorption capacity, but also the CO_2_ photocatalytic conversion performance, which may be due to the coupling effect between them. In other words, the CO_2_ consumption by the reduction reaction promoted the adsorption equilibrium to move forward. Simultaneously, the strong adsorption increased the concentration of CO_2_ on the material surface, especially on the active site of Cu^2+^, thus promoting the catalytic reduction reaction.

## 4. Conclusions

Amine-modified photocatalysts Cu-Mg-Al LDHs have been prepared using N-aminoethyl-γ-aminopropyltrimethoxysilane as a grafting agent. The CO_2_ adsorption capacity of Cu-Mg-Al N was up to 1.82 mmol·g^−1^ at 30 °C and a 0.1 MPa pure CO_2_ atmosphere. The amine-modified method and an appropriate Cu^2+^ content have great influences on the surface morphology of the materials, and promising amine loadings and the surface free-amino functional groups were obtained. The kinetic model confirmed that the complex CO_2_ adsorption mechanism of Cu-Mg-Al LDHs involves a rapid phase controlled by chemical adsorption with strong surface reactions and a slow phase controlled by physical adsorption with intraparticle diffusion. Both of them were enhanced with the increase of the Cu^2+^ content, while chemical adsorption was suppressed when the Cu^2+^ content was too high. A slight decrease in the adsorption capacity at high temperatures and excellent adsorption performance stability throughout three sequential adsorption-regeneration cycles of Cu-Mg-Al N provided theoretical supports of adsorption for its further application to CO_2_ photocatalytic conversion. Cu-Mg-Al N can photocatalytically reduce CO_2_ to methanol with Cu^2+^ as an active site, which fully demonstrated its potential in the fields of photocatalysis and adsorption of flue gas from power plants.

## Figures and Tables

**Figure 1 nanomaterials-12-00127-f001:**
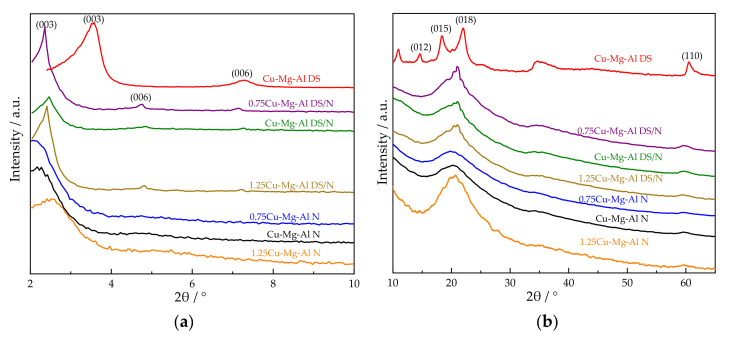
XRD diffraction patterns of Cu-Mg-Al DS, xCu-Mg-Al DS/N, and xCu-Mg-Al N at different 2*θ* ranges: (**a**) 2*θ* range of 2°–10°; (**b**) 2*θ* range of 10°–65°.

**Figure 2 nanomaterials-12-00127-f002:**
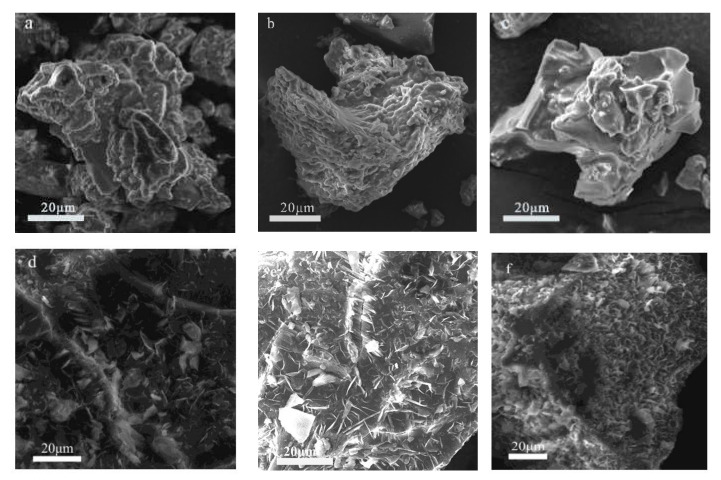
SEM micrographs of the samples: (**a**) 0.75Cu-Mg-Al N; (**b**) Cu-Mg-Al N; (**c**) 1.25Cu-Mg-Al N; (**d**) 0.75Cu-Mg-Al DS/N; (**e**) Cu-Mg-Al DS/N; (**f**) 1.25Cu-Mg-Al DS/N.

**Figure 3 nanomaterials-12-00127-f003:**
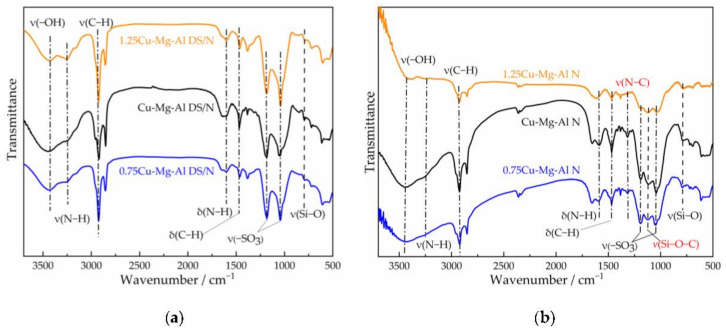
FTIR spectra of the samples: (**a**) xCu-Mg-Al DS/N; (**b**) xCu-Mg-Al N.

**Figure 4 nanomaterials-12-00127-f004:**
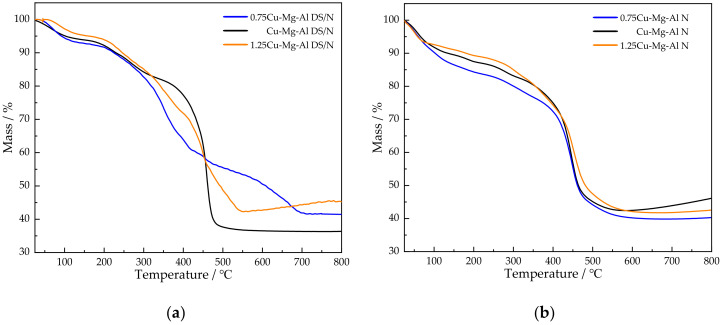
Thermogravimetric (TG) curves of the samples: (**a**) xCu-Mg-Al DS/N; (**b**) xCu-Mg-Al N.

**Figure 5 nanomaterials-12-00127-f005:**
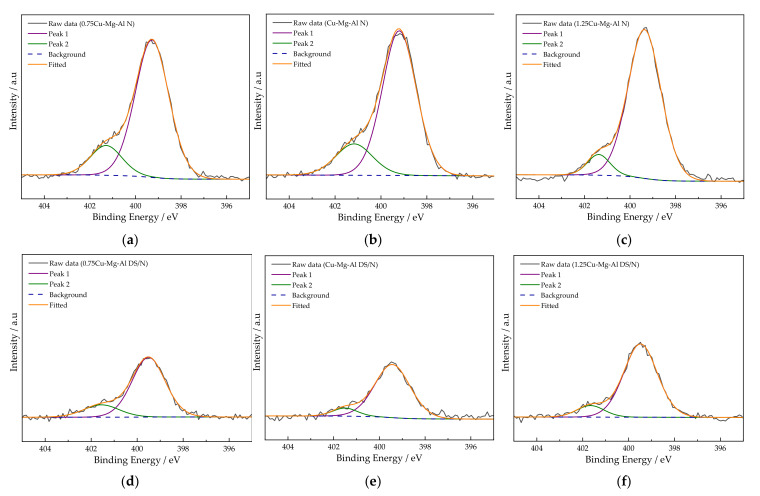
The high-resolution spectra of N 1s of xCu-Mg-Al DS/N and xCu-Mg-Al N: (**a**) 0.75Cu-Mg-Al N; (**b**) Cu-Mg-Al N; (**c**) 1.25Cu-Mg-Al N; (**d**) 0.75Cu-Mg-Al DS/N; (**e**) Cu-Mg-Al DS/N; (**f**) 1.25Cu-Mg-Al DS/N.

**Figure 6 nanomaterials-12-00127-f006:**
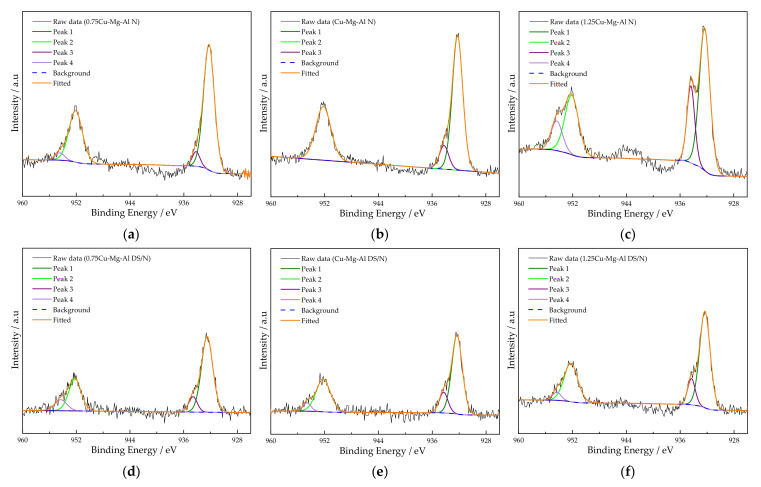
The high-resolution spectra of Cu 2p of xCu-Mg-Al DS/N and xCu-Mg-Al N: (**a**) 0.75Cu-Mg-Al N; (**b**) Cu-Mg-Al N; (**c**) 1.25Cu-Mg-Al N; (**d**) 0.75Cu-Mg-Al DS/N; (**e**) Cu-Mg-Al DS/N; (**f**) 1.25Cu-Mg-Al DS/N.

**Figure 7 nanomaterials-12-00127-f007:**
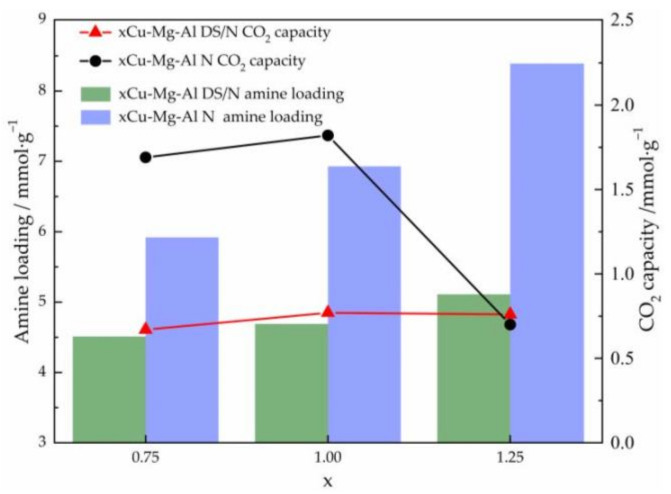
Relationships between the CO_2_ adsorption capacity and the amine loading content for xCu-Mg-Al DS/N and xCu-Mg-Al N.

**Figure 8 nanomaterials-12-00127-f008:**
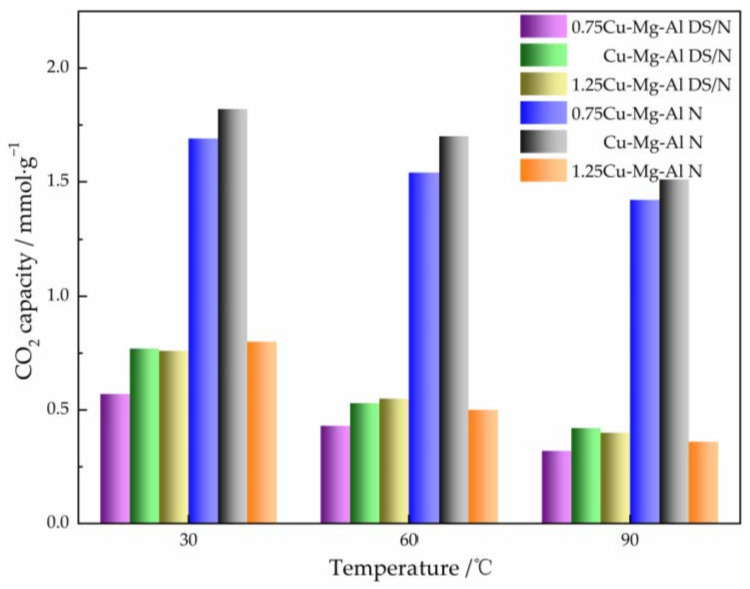
CO_2_ adsorption capacities of xCu-Mg-Al N and xCu-Mg-Al DS/N with different temperatures.

**Figure 9 nanomaterials-12-00127-f009:**
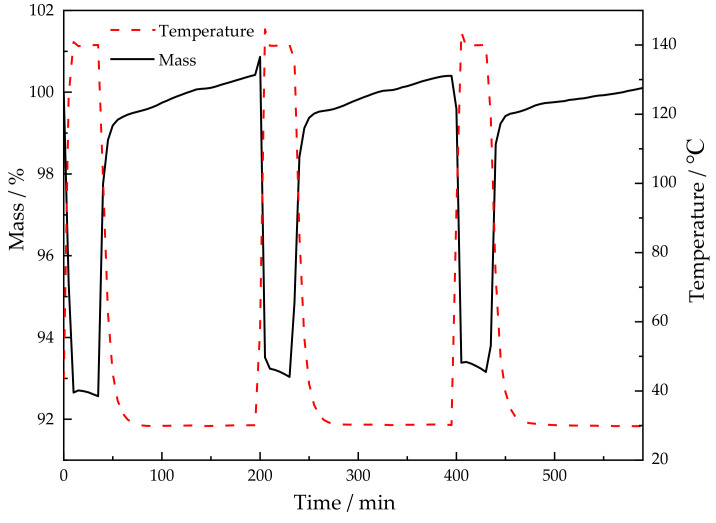
Adsorption-regeneration cycling results of Cu-Mg-Al N.

**Figure 10 nanomaterials-12-00127-f010:**
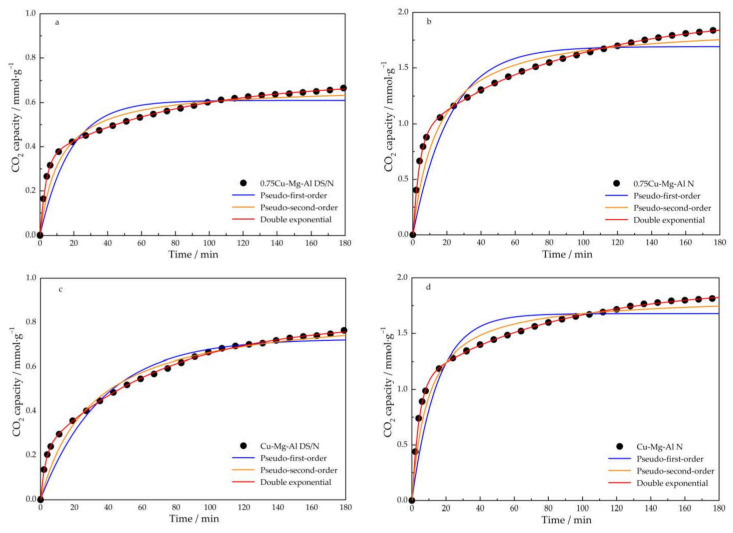
Comparison of different kinetic models with the experimental data of the CO_2_ adsorption on 0.75Cu-Mg-Al DS/N (**a**), 0.75Cu-Mg-Al N (**b**), Cu-Mg-Al DS/N (**c**), Cu-Mg-Al N (**d**), 1.25Cu-Mg-Al DS/N (**e**), and 1.25Cu-Mg-Al N (**f**).

**Table 1 nanomaterials-12-00127-t001:** Elemental analysis results of xCu-Mg-Al DS/N and xCu-Mg-Al N.

Sample	Elemental Weight (%)				Formula for Intercalated Molecules	Amine Loading
	N	C	H	S	(mmol·g^−1^)	(mmol·g^−1^)
0.75Cu-Mg-Al DS/N	6.31	36.58	7.65	3.95	(C_12_H_25_SO_3_^−^)_1.24_(C_6.95_H_20.23_SiN_2_O_3_)_2.25_	4.51
Cu-Mg-Al DS/N	6.57	36.62	7.71	4.47	(C_12_H_25_SO_3_^−^)_1.40_(C_5.87_H_17.96_SiN_2_O_3_)_2.35_	4.69
1.25Cu-Mg-Al DS/N	7.15	36.27	7.53	4.28	(C_12_H_25_SO_3_^−^)_1.34_(C_5.55_H_16.39_SiN_2_O_3_)_2.55_	5.11
0.75Cu-Mg-Al N	8.29	34.34	7.42	2.04	(C_12_H_25_SO_3_^−^)_0.64_(C_7.08_H_19.68_SiN_2_O_3_)_2.96_	5.92
Cu-Mg-Al N	9.70	36.30	7.49	1.49	(C_12_H_25_SO_3_^−^)_0.47_(C_7.12_H_18.26_SiN_2_O_3_)_3.46_	6.93
1.25Cu-Mg-Al N	11.75	35.98	7.83	1.46	(C_12_H_25_SO_3_^−^)_0.46_(C_5.84_H_15.94_SiN_2_O_3_)_4.20_	8.39

**Table 2 nanomaterials-12-00127-t002:** Weight loss ratios of xCu-Mg-Al DS/N and xCu-Mg-Al N.

Sample	First Stage (%)	Second Stage (%)	Third Stage (%)
0.75Cu-Mg-Al DS/N	7.7	8.5	43.2
Cu-Mg-Al DS/N	6.1	9.7	47.8
1.25Cu-Mg-Al DS/N	4.8	10.1	39.9
0.75Cu-Mg-Al N	13.5	6.5	39.7
Cu-Mg-Al N	10.5	6.3	37.0
1.25Cu-Mg-Al N	8.9	6.1	42.4

**Table 3 nanomaterials-12-00127-t003:** CO_2_ adsorption capacities for the amine-modified layered double hydroxides (LDHs), as cited in the literature.

Sample	Grafting Agent	Temperature (°C)	CO_2_ Adsorption (Capacity mmol^−1^·g^−1^)	Reference
Mg-Al N2	N-aminoethyl-γ-aminopropyltrimethoxysilane	30	2.26	[26]
NiMgAl N2	N-[3-(Trimethoxysilyl)propyl]ethylenediamine	80	2.02	[27]
Cu-Mg-Al N	N-aminoethyl-γ-aminopropyltrimethoxysilane	30	1.82	This work
MgAl N3	3-[2-(2-Aminoethylamino) ethylamino]propyl-trimethoxysilane	80	1.76	[28]
UL30-LDH	N-aminoethyl-gamma-aminopropyltrimethoxysilane	30	1.65	[56]
MgAl MEA 1	3-aminopropyl triethoxysilane	25	1.39	[29]
UH-MEA5	(3-aminopropyl)-triethoxysilane	55	1.37	[57]
N1-HMS@Mg-Al LDH	N1-(3-Trimethoxysilylpropyl) diethylenetriamine	75	1.28	[23]

**Table 4 nanomaterials-12-00127-t004:** Parameters of CO_2_ kinetic models, R^2^, and standard errors (%) for xCu-Mg-Al N and xCu-Mg-Al DS/N at 30 °C and 1 atm.

Samples	Pseudo-1st-Order		Err	R^2^	Pseudo-2nd-Order		Err	R^2^	Double-Exponential					Err	R^2^
	q_e_	k_f_	(%)		q_e_	k_f_	(%)		q_e_	A1	A2	k1	k2	(%)	
0.75Cu-Mg-Al DS/N	0.61	0.06	0.45	0.8139	0.67	0.13	0.43	0.9372	0.70	0.35	0.36	0.01	0.31	0.18	0.9989
Cu-Mg-Al DS/N	0.72	0.02	0.56	0.9357	0.86	0.03	0.71	0.9718	0.80	0.59	0.22	0.01	0.37	0.18	0.9993
1.25Cu-Mg-Al DS/N	0.69	0.03	0.62	0.8675	0.80	0.06	0.74	0.9438	0.88	0.55	0.31	0.01	0.20	0.42	0.9994
0.75Cu-Mg-Al N	1.69	0.04	1.31	0.8409	1.89	0.03	1.35	0.9434	1.97	1.07	0.94	0.01	0.27	0.72	0.9987
Cu-Mg-Al N	1.67	0.07	1.11	0.8270	1.83	0.05	0.97	0.9497	1.92	0.86	1.11	0.01	0.26	0.68	0.9983
1.25Cu-Mg-Al N	0.73	0.01	0.82	0.9772	0.99	0.01	1.25	0.9866	1.01	0.86	0.15	0.01	0.12	0.62	0.9999

**Table 5 nanomaterials-12-00127-t005:** Product chromatographic results for different photocatalysts.

Catalyst	Product	Retention Time (min)	Content (%)
Mg-Al N	CO_2_	0.473	4.30
	H_2_	0.973	95.40
Cu-Mg-Al DS/N	CO_2_	0.520	3.76
	H_2_	1.020	83.08
	CH_3_OH	3.850	12.40
Cu-Mg-Al N	CO_2_	0.530	3.43
	H_2_	1.030	77.24
	CH_3_OH	3.840	18.53

## Data Availability

The data presented in this study are available from the corresponding author on request.

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
