# Peer review of "Preparation of Amine-Modified Cu-Mg-Al LDH Composite Photocatalyst"

_nanomaterials, 2021, doi:10.3390/nano12010127_

Round 1

Reviewer 1 Report

The manuscript “Preparation of composite amine modified photocatalyst Cu-Mg-Al LDH” by Qining Wang et al.,reports on the synthesis of amines-modified Cu-Mg-Al LDHs and studied for CO2 photocatalytic reduction and their adsorption capacities.

After a careful reading, I suggest that major revisions are highly needed. Few questions and remarks are listed here below.

Paper Title: The title should be modified such as: Preparation of amine-modified Cu-Mg-Al LDH composite photocatalyst

  1. Introduction

The entire Introduction section needs to be revised. The references should be updated and more recent paper focusing on LDHs as photocatalysts need to be cited. Too long sentences in introduction (even 5-6 lines!) which are very difficult to follow; therefore I suggest shortening them.

The aim of the paper it is not very clear explained. The sentence “The materials were characterized using EA, XDR, SEM. FTIR, TG and XPS measurements techniques etc…. ” is repeated, identically idea was expressed in Abstract section. Please revise.

2. Materials and Methods

Subtitle 2.2.

The two sentences “Dispersion of 1.0 g of xCu-Mg-Al DS/N in 100 mL toluene etc……” and “The mixture was stirred at 70 ⁰C for 20 h etc…” needs to be reformulated by avoiding the repetition of the same words.

It is not adequate to use the expression “Varying Cu2+doping content, three samples were produced etc…..” The quantities of Cu2+are too high, so this is a synthesis of LDHs with 3 cationic species (Cu2+, Mg2+ and Al3+) and not a Cu doping.

Subtitle 2.5. Which  type of Xe lamp has been used? It was a short-arc lamp? Which is the power lamp?  Which  type of  column chromatograph ? Please details.

3. Results and Discussion

Subtitle 3.1.1.: The paragraph “The amine loadings of xCu-Mg-Al N after composite amine modified are increased significantly…..is smaller than that of magnesium” is not clearly explained. Please revise.

Subtitle 3.1.2.Fig 1 The legend of Figure 1 is not clear; please explain the differencesbetween figures (a) and (b). The XRD patterns of 0.75Cu-Mg-Al DS/N and 1.25Cu-Mg-Al DS/N are missing. Also, the un-modified LDH solid is missing. Please give it and explains.

Regarding all characterization figures, if you decided to give black graphs please differentiate the samples by straight lines, dotted lines, etc. It is impossible to distinguish between samples.

Subtitle 3.1.3.SEM image of Cu-Mg-Al DS/N must have the same size scale as the other samples. Please revise.

Also, TEM images would be useful.

Subtitle 3.1.4. Please give the FTIR spectra of 0.75Cu-Mg-Al DS/N, 1.25Cu-Mg-Al DS/N and Cu-Mg-Al DS samples and details.

Please, also provide the TG curves of all xCu-Mg-Al DS/N and reformulate the discussion on thermal analysis by comparing both series of the synthesized samples.

Subtitle 3.1.6. How Cu2+insertion in the MgAl LDH influence the final architecture of the composite materials? Only N1s spectrum is shown, please give the Cu2p XPS spectra of samples.

Again, which in the reason to characterized by XPS only the Cu-Mg-Al DS/N from this series of synthesis? Why not all of them as in the case of xCu-Mg-Al N ones?

In the legend of Table 3 is mentioned the “surface content” while the values form the table gives the areas of free and protonated amines. In fact, these are the intensities of XPS peaks which in deed demonstrate the increase/decrease of amine groups. Please reformulate.

3.2.CO2 adsorption performance: Which are the specific surface area of synthesized LDH composite photocatalystsand how does influences the CO2 adsorption?

The specific surface area of synthesized LDH would be useful

3.3 The gases by CO2 reduction ….the sentence can be reformulate.

It would also be useful to show how content of Cu2+influences the catalytic efficiency in photocatalytic CO2 reduction.  

Reviewer 2 Report

No comments

Author Response

Thank you.

Reviewer 3 Report

According to the XRD patterns, most of the samples have lost the layered structure and is not layered anymore, however, the authors are still claiming the structures are layered and LDH. they should be more studied and reconsidered. 

The LDH structures are unique but most of the times are unstable, therfore, in the introduction, it should be noted that why are unstable and how they can be stabilized, such as post-modifications or intercalation with other stabilizing agents, I recomemend the use of the follwing reference: Tahawy et al., https://doi.org/10.1016/j.apcatb.2020.119854

Furthermore, the LDHs can be used as MOF precursors and play as a sacrifical template for the synthesis of MOFs. See the follwing paper which is published in within this year in Chemical Socity Reviews, volume 50, page 2927.

CO2 adsorption is not compared with otehr materials and needs comparision with other materials. 

Round 2

Reviewer 1 Report

I appreciate the authors' revision, which completed all my concerns. I have only two remarks:

1) The paragraph 3.2. CO2 Adsorption Performance from page 10 is superposed with the legend of Figure 6. Please, revise this typing error.

2). Please pay attention on Figure 10 from page 13 where the graphs overlapped.

Reviewer 3 Report

The paper is revised accordingly and can be accepted in the journal!

This manuscript is a resubmission of an earlier submission. The following is a list of the peer review reports and author responses from that submission.